# Chitosan Functionalized with 2-Methylpyridine Cross-Linker Cellulose to Adsorb Pb(II) from Water

**DOI:** 10.3390/polym13183166

**Published:** 2021-09-18

**Authors:** Jorge Lozano-Montante, Raquel Garza-Hernández, Mario Sánchez, Edgar Moran-Palacio, Guillermo Niño-Medina, Mario Almada, Luis Hernández-García

**Affiliations:** 1Centro de Investigación e Innovación Tecnológica, Tecnológico Nacional de México/IT Nuevo León, Av. de la Alianza No. 507, PIIT, Carretera Monterrey-Aeropuerto Km. 10, Apodaca 66628, Nuevo León, Mexico; jorgelm_22@outlook.com; 2Centro de Investigación en Materiales Avanzados, Alianza Norte 202, Parque de Investigación e Innovación Tecnológica, Apodaca 66628, Nuevo León, Mexico; raquel.garza@cimav.edu.mx (R.G.-H.); mario.sanchez@cimav.edu.mx (M.S.); 3Departamento de Ciencias Químico-Biológicas y Agropecuarias, Universidad de Sonora, Lázaro Cárdenas 100, Colonia Francisco Villa, Navojoa 85880, Sonora, Mexico; edgar.moran@unison.mx; 4Laboratorio de Química y Bioquímica, Facultad de Agronomía, Universidad Autónoma de Nuevo León, Francisco Villa S/N, Col. Ex-Hacienda El Canadá, General Escobedo 66050, Nuevo León, Mexico; guillermo.ninomd@uanl.edu.mx

**Keywords:** adsorption, isotherms, kinetics, chitosan, cellulose

## Abstract

In this study, chitosan was chemically modified with 2-methylpyridine. Subsequently, the modified chitosan was cross-linked to cellulose using succinic anhydride. Additionally, the capacity of cellulose derivatives to adsorb Pb(II) ions in an aqueous solution was studied through the determination of Pb(II) ions concentration in water, using microwave plasma atomic emission spectroscopy (MP-AES). A maximum adsorption capacity of 6.62, 43.14, 60.6, and 80.26 mg/g was found for cellulose, cellulose-succinic acid, cellulose-chitosan, and cellulose-chitosan-pyridine, respectively. The kinetic data analysis of the adsorption process showed a pseudo-second-order behavior. The increase in metal removal from water is possibly due to metal chelation with the carbonyl group of succinic acid, and the pyridine groups incorporated into chitosan.

## 1. Introduction

Anthropogenic activity results in the generation of different wastes; in many cases, these are toxic to humans and the ecosystem. One of the most critical problems in this regard is water contamination by heavy metallic ions. Despite being in very low concentration, these pollutants could seriously damage ecosystems and human health. Heavy metals usually have densities greater than 5 g/cm^3^; the most toxic metals, lead (Pb), cadmium (Cd), and mercury (Hg), which are not biodegradable, could be bioaccumulated, causing health problems in living systems [1,2,3]. For example, recent research shows that lead solubility increases in simulated acid rain because of alterations in plant growth [4,5]. For this reason, the Agency for Toxic Substances and Disease Registry considers lead to be the most hazardous of all harmful metals, only after arsenic [6,7].

Lead is found in the soil of the earth’s crust, much like sulfates, carbonates, or oxides [8]. It is a malleable metal located in group 14 of the periodic table, with a common oxidation state of +2 (Pb(II)) [9]. There is no such thing as a nontoxic level of lead in the human body, so in very small quantities, lead causes serious health problems in developing countries due to a lack of strict regulations. Recently, Tamayo and Ortiz et al. reported that some candies sold in Mexico City are contaminated with lead, and are mainly consumed by children [10]. Although lead concentrations in most candies are within the limits allowed by the FDA, high percentages of around 12% lead have been found in candy brands that were analyzed by inductively coupled plasma mass spectrometry (ICP-MS). Another source of lead contamination is the use of lead-acid batteries in household appliances and car batteries. In these cases, there is the risk of environmental contamination beginning with the manufacturing of the product, all the way to the end of its functional life due to a lack of adequate control [11,12]. In this sense, the pollution increases each year as more lead is released into the environment [13]. Although the concentration of this metal is very low, it can still cause severe neurological, reproductive, and cardiovascular problems in humans, and growth problems in children [14].

There are different methods used to separate heavy metals from water, such as precipitation, ion exchange, and adsorption. The adsorption process is used in different fronts due to its technical versatility and cost effectiveness. However, this process is limited because it does not show an acceptable effectiveness at pH levels below four. This is a consequence of active sites that are rich in electron pairs; these areas can be occupied by an excess of protons, which generates competition in the active sites. If this occurs, it results in a decrease in adsorption [15,16,17]. In this context, the development and improvement of materials that can avoid contamination or remove metals is a topic that continues to attract the attention of different research groups [18,19,20,21,22]. The use of biomaterials is frequently reported in the literature; they are low-cost, and some of them are effective materials for metal adsorption. Furthermore, they could potentially be modified and functionalized for greater efficiency [23,24,25]. The two most studied biopolymers that remove heavy metals from water are chitosan and cellulose [26,27]. Zhang et al. realized oxidation in cotton fiber with NaIO_4,_ carrying out a Schiff-base-type chitosan reaction between amino groups in the chitosan and the aldehyde group generated in the cotton fiber [28]. Xiao and Hu reported the use of cellulose and chitosan derivatives to adsorb heavy metal ions. They used a basic-based solvent system to prepare the derivatives, and reported an adsorption capacity of 0.16 mmol/g of Pb(II) ions at pH 5 [29].

Recently, our investigation group published the results of cross-linking cellulose-cotton with chitosan using green methodology, with moderate capacity adsorption [30]. Other research groups have reported that chitosan is widely modified to improve its adsorption capacity. One of these modifications involves the incorporation of pyridine derivatives [31,32,33,34]. In this context, the present study shows the results of the synthesis of cellulose cross-linking with a chitosan-pyrimidine derivative using a fast and accessible methodology. This modification produces more sites that can interact with lead, such as carbonyl and chitosan-picolyl groups, which is expected to make the adsorption and transport of the metal in water more efficient. In addition, the kinetics of removing Pb(II) from an aqueous solution was studied, and results showed that this modification increases the adsorption capacity to pH 5.

## 2. Materials and Methods

### 2.1. Starting Materials

All reagents were from Sigma Aldrich (St Louis, MO. USA), including chitosan (CS), which showed a deacetylation degree of ≥75%, and a molecular weight of around 150 kDa. Raw cellulose fiber in the form of commercial cloth was donated from the Parisina store (Monterrey, Mexico). The cellulose was washed with neutral soap and later washed with 0.1 M of HCl solution for 5 min, then washed again with distilled water.

### 2.2. Characterization Materials

Samples were analyzed with a Nicolet iS10 FT-IR spectrometer (Thermo Scientific, Waltham, MA, USA) using the ATR technique in the region of 4000–600 cm^−1^. All surface images of the polysaccharide derivative material and elements on the surface were determined by a scanning electron microscope (SEM) Nova NanoSEM200 (FEI, Hillsboro, OR, USA) using a low vacuum detector (LV). We recorded x-ray photoelectron spectroscopy (XPS) with an Escalab 250Xi Thermo Scientific (Santa Clara, CA, USA); the spectra were recorded at a take-off angle of 90° and monochromatic Al Kα, from at least three locations on each sample, in 4.3 eV steps with an analyzer pass energy of 50 eV.

### 2.3. Preparation of 2-Pyridyl Methyl Chitosan (Cs-Py)

For the synthesis of the 2-pyridyl methyl chitosan derivative (Cs-Py), 1.0 g of chitosan and 1.0 g of 2-(chloromethyl) pyridine hydrochloride were mixed into 10 mL of NaOH 0.1 M and left to stir for 24 h. The chitosan derivative was washed with 100 mL water, and later with 50 mL of ethanol. Finally, Cs-Py was dried at room temperature.

### 2.4. Preparation of Cellulose-Succinic (Com-1)

To obtain the cellulose-succinic (Com-1) derivative, 1.0 g of succinic anhydride and 1.0 g of cellulose were placed into 60 mL of dry pyridine. The mixture was left under reflux for 12 h under a nitrogen atmosphere. The cellulose was then washed with 200 mL of deionized water and 50 mL of acetone. Finally, the cellulose-succinic (Com-1) derivative was left to dry at room temperature.

### 2.5. Preparation of Cellulose-Succinic-CDI (Com-2)

For the preparation of the cellulose-succinic-CDI (Com-2), 1 g of Com-1 and 0.5 g of 1,1′-carbonyl-diimidazol (CDI) were placed in a flask with 100 mL of DMF. The mixture was stirred at room temperature for 2 h. The cellulose fibers were then washed with H_2_O (50 mL) and acetone (20 mL), and the derivative was left to dry at room temperature.

### 2.6. Preparation of Cellulose-Succinic-Chitosan (Com-3) and Cellulose-Succinic-Chitosan-Pyridine (Com-4)

For the preparation of cellulose-succinic-chitosan (Com-3), 1.0 g of chitosan was placed into 100 mL of HCl solution 0.1 M. The mixture was stirred until dissolution occurred, and the pH was adjusted to 5 with NaOH. Subsequently, Com-2 was added to the solution, and the reaction mixture was left under reflux overnight. Cellulose-succinic-chitosan (Com-3) was washed with 50 mL of HCl 0.1 M to eliminate free chitosan. The product was then dried at room temperature. Later, Com-3 was washed with 50 mL of 0.1 M NaOH and 50 mL of water. Finally, the derivative was dried at room temperature. Synthesis of cellulose-succinic-chitosan-pyridine (Com-4) was realized under the same conditions as Com-3, exchanging chitosan for Cs-Py.

### 2.7. Adsorption Kinetics

To evaluate Pb(II) removal from water, a batch adsorption set-up was employed. To analyze adsorption kinetics on the cellulose cross-linked Com-3 and Com-4, we prepared a stock solution of 100 ppm of Pb(II) by dissolving analytical grade Pb(NO_3_)_2_ in 3% HNO_3_ at pH 5. We carried out experiments to determine adsorption capacity at 150 rpm in an incubated Benchtop Orbital Shaker (Thermo Scientific SHKA4000-5). Cellulose, Com-1, Com-3, or Com-4 were dried overnight at room temperature. After that, 0.5 g of the dried derivatives was placed in 100 mL of 100 ppm Pb(II) solution and left to stir for 24 h. After 0.5, 1, 2, 3, 4, 5, 6, and 24 h, an aliquot of 2 mL was taken, and the Pb(II) concentration was measured with an Agilent MP-AES 4200 microwave plasma-atomic emission spectrometer (Santa Clara, CA, USA) using a wavelength of 405.78 nm and a nebulizer pressure of 0.95 L/min. The adsorption capacity *q_t_* (mg/g) was calculated based on the difference between the Pb(II) concentration in the aqueous solution before and after adsorption according to the formula:
(1)$$q_{t}=\frac{V\left(C_{0}-C_{t}\right)}{m}$$
where *C*_0_ and *C_t_* (mg/L) are the adsorbate concentration at the initial time (0) and any given time (*t*, in min), respectively; *V* (L) is the volume of the adsorbate solution; *m* (g) is the mass of the adsorbents; and *q_t_* (mg/g) is the adsorbed amount at time *t* (adsorption capacity). Additionally, experimental data were adjusted to pseudo-first- and pseudo-second-order kinetic models to determine the rate-controlling step [35]. Each graph corresponds to the averege of three independent experiments, and the Pb(II) concentration measurement in each experiment was performed in triplicate.

Pseudo-first-order kinetic model:
(2)$$ln\left(q_{e}-q_{t}\right)=lnq_{e}-k_{1}t$$

Pseudo-second-order kinetic model:
(3)$$\frac{t}{q_{t}}=\;\frac{1}{k_{2}q_{e}^{2}}+\frac{t}{q_{e}}$$

### 2.8. Adsorption Isotherms

Adsorption isotherms describe the relationship between adsorbents and adsorbates when equilibrium is attained. The adsorption isotherms of Pb(II) on cellulose derivatives were obtained by adding 0.5 g of cellulose, Com-1, Com-3, or Com-4 into 100 mL of heavy metal ion solution, with concentrations ranging from 12.5 to 400 mg·L^−1^. All adsorption isotherm experiments were realized at a pH of 5. Dispersions were shaken mechanically at 150 rpm in room temperature, and 1 mL of the solution was extracted after 24 h. The Pb(II) concentration was measured with an Agilent MP-AES 4200 microwave plasma-atomic emission spectrometer (Santa Clara, CA, USA) using a wavelength of 405.78 nm, and a nebulizer pressure of 0.95 L/min. Experimental data were fitted using Langmuir and Freundlich isotherm models:

Langmuir model:


(4)
$$\frac{C_{e}}{q_{e}}=\;\frac{1}{q_{\;m}K_{L}}+\frac{C_{e}}{q_{\;m}}$$


Freundlich model:
(5)$$logq_{e}=\;logK_{F}+\frac{logC_{e}}{n}$$
where *q_e_* (mg/g) is the amount of Pb(II) adsorbed onto the cellulose derivative at equilibrium; *q_m_* (mg/g) is the saturated adsorption capacity of Pb(II) adsorbed onto the cellulose derivative to form a complete monolayer coverage; *K_L_* (L/mg) is the Langmuir adsorption coefficient; *K_F_* is the Freundlich adsorption coefficient related to the adsorption capacity; *n* is an indicator of isotherm nonlinearity corresponding to the adsorption intensity at specific temperatures [36]. Each graph corresponds to the averege of three independent experiments, and the Pb(II) concentration measurement in each experiment was performed in triplicate.

## 3. Results

Carboxylic and pyridine are potential functional groups that increase the ability of cellulose and chitosan to form ionic complexes with heavy metals in water. In this study, a chitosan chemical modification was conducted by adding 2-bromomethyl pyridine at a 1:1 weight ratio, resulting in the chitosan derivative chitosan-pyridine (Cs-Py) (Figure 1, Figure 1a). Cellulose fibers were modified using carboxylic groups (Com-1) through a reaction between the succinic anhydride and C6 hydroxyl groups in the cellulose molecules. Once washed with water and acetone, Com-1 was reacted with CDI in DMF to obtain its corresponding ester.

To continue the cellulose functionalization, we cross-linked Com-2 with chitosan or Cs-Py using DMSO as solvent at reflux temperature generating the derivatives Com-3 and Com-4, respec-tively (Figure 2).

## 4. Characterization

### 4.1. Fourier Transform Infrared (FTIR) Analysis

The FTIR technique was employed to confirm the reaction between chitosan and methyl pyridine. Figure 1a shows the chitosan spectrum, which reveals the characteristic peaks for this polymer. The hydroxyl group (–OH) stretching vibration and the amine (–NH) asymmetric vibration appear at 3357 and 3289 cm^−1^, respectively. Vibrations at 1644 and 1588 cm^−1^ were attributed to a carbonyl group (C=O) stretching vibration, related to incomplete chitosan deacetylation. Figure 1b shows the chitosan-pyridine spectrum, where we observe the characteristic peaks for chitosan at 1651 and 1595 cm^−1^. Additionally, new absorption peaks at 1570 and 763 cm^−1^ arise, which were assigned to amide and pyridine groups, respectively.

To confirm the functionalization of cellulose, the FTIR technique was also used; results for Com-1, Com-3, and Com-4 derivatives are shown in Figure 2. The cellulose spectrum shows a strong absorption peak at 3332 cm^−1^, which is characteristic of –OH stretching vibration. Additionally, the band at 2898 cm^−1^ can be attributed to anti-symmetric stretching for C–H onto the cellulose structure. The bands at 1159 cm^−1^ and 1108 cm^−1^ were associated with the ring stretching vibration of the C–C bond and the C–O–C bond of the glyosidic bonds, respectively. A band observed at 898 cm^−1^ was attributed to β-glycosidic linkage.

For all derivatives, the signals described for cellulose appear at similar wavelengths; however, when new peaks appeared, they were related to functionalization. Com-3 shows a strong absorption band at 1697 cm^−1^, corresponding to carbonyl group C=O stretching vibration due to the incorporation of the succinate structure. We attributed signals at 1556 cm^−1^ for Com-3 and 1559 cm^−1^ for Com-4 to carbonyl group C=O stretching vibration due to the presence of amide bonds [37].

Figure 3 shows the FTIR comparison for the Com-4 composite before and after Pb(II) adsorption. A decrease in the intensity of several signals (1685, 1559, 1427, 1105, and 895 cm^−1^) due to adsorption of lead by the composite can be observed (see also Appendix A). This behavior was reported previously, and was attributed to an interaction between nitrogen atoms and lead ions. This suggests that Pb(II) adsorption in Com-4 is, at least, partially carried out by amino groups from chitosan [38,39,40]. Figure 3 shows the composite with a possible coordination site for the lead atom, where nitrogen atoms and the carbonyl group form interactions with the metal. According to Pearson’s concept of hard and soft acids and bases, the composite could behave as a hard base due the presence of electron pairs in the nitrogen and oxygen atoms, which interact preferentially with metal ions, such as Pb(II), which is considered a borderline atom [41,42].

### 4.2. Surface Analysis by XPS

Figure 4 shows the high-resolution spectra corresponding to the C 1*s*, N 1*s*, Pb 4*f*, and O 1*s* core energetic levels of cellulose, Com-1, Com-3, and Com-4 derivatives. The C 1*s* spectra region includes four components. The first peak located at 284.8 eV was attributed to C–H/C–C chemical bonds; the C–NH_2_ bond can be overlapped by this peak. The second peak is centered at 286.4 eV, which was assigned to C–OH, C–O, and C–N bonds. The third peak found at 287.9 eV corresponded to O–C–O/N–C=O bonds. Finally, the peak at 289.0 eV was associated with the C=O or O=C–O bonds [43,44,45,46].

The N 1*s* region showed different components depending on the sample. The nitrogen signal was observed in cellulose and Com-1 samples, even though they do not contain the element in their structures. This impurity could have been adsorbed into these samples from the atmosphere. Furthermore, we observed four peaks for Com-3 and Com-4 samples. The peak located at the higher binding energy (401.3 eV) could be for protonated amines [46,47].

The protonation of amines likely occurred due to the Pb(II) adsorption reaction, which was carried out in a slightly acidic medium. The contribution of –NH–C=O bonds is seen in the peak centered at 400.1 eV. These bonds correspond to the chemical interaction between amine groups, from chitosan and succinic acid, and cellulose. The peak found at 399.3 eV is related to the non-protonated amine from chitosan [46,47]. Particularly, the component at the lower binding energy value (398.6 eV) was associated with the –N=CH_2_– bond from the pyridine molecule in the case of the Com-4 sample, and to the –N=CH_2_–N–R_2_ bond from imidazole prevails for Com-3 sample [48,49,50].

The Pb 4*f* region constitutes Pb 4*f_7/2_* and Pb 4*f_5/2_* core level peaks. The doublet peaks centered at 138.9 and 140.1 eV, with a spin-orbit splitting of 4.84 eV, are associated with Pb–O and Pb–NO_3_ bonds [51]. The oxygen atoms in the structures of different derivatives promote the adsorption of lead due to its electronegativity. Although Pb can coordinate with pyridine, there are no reports mentioning its binding energy. The oxidation state of Pb in these samples was +2.

Finally, the spectra for O1*s* are constituted by three peaks. The peak located at the lower binding energy value (531.3 eV) was attributed to O–C bonding, which is formed due to the adsorption of CO_2_ from the atmosphere; the peak at 532.7 was assigned to H–O–C bonds [44,45]. The peak centered at the higher binding energy value (534.2 eV) was associated with –O–C=O bonds. These bindings come from succinic acid [52]. The bond O–Pb is not easy to distinguish because it is overlapped by the O–C peak at 531.4 eV.

Figure 5 shows the high-resolution spectra of composites before lead adsorption. The main differences between the spectra before and after lead adsorption can be observed in Com-3 and Com-4 composites. Essentially, the atomic ratio of species C–OH, C–O/C–C, and C–H before lead adsorption was higher than those that come after, and higher than the cellulose and Com-1 spectra. This result was expected due to the high molecular weight of the composites compared to the cellulose and Com-1. There was a high content of bonds C–OH and C–O in these structures. The reduction of these functional groups after lead adsorption was related to the signal attenuation by lead. (In Appendix A shows the atomic concentration of different functional groups.)

Table 1 shows the percentages of the relative atomic concentrations of all elements found in the samples. The derivatives contain low concentrations of silicon, which could be considered contamination. The theoretical atomic ratios of O/C for cellulose, Com-1, Com-3, and Com-4 are 0.83, 0.78, 0.79, and 0.64, respectively. However, the atomic ratio of those derivatives was observed at around 0.44–0.47. The O/C atomic ratio reduction may be related to the reaction in the hydroxyl groups to the derivatives, with the acidic medium used for the adsorption of Pb forming H_2_O as a subproduct. The atomic concentration (%) of the C–C/C–H and O=C–O/C=O of Com-1 was higher than that of cellulose. This increase in atomic concentrations can be explained by the addition of succinic acid, which was induced by cross-linking reactions. The cross-linking with chitosan and pyridine reduced the atomic concentration of these groups due to the increase in –NH_2_ and N-pyridine bonds. In contrast, the Com-3 sample had the highest content of Pb adsorbed at the surface. The discrepancy observed between the results of the XPS analyses and adsorption kinetics was due to the volume in each technique.

### 4.3. Surface Analysis by SEM and EDS

Scanning electron microscopy images and energy-dispersive x-ray spectra from cellulose and cellulose derivatives after Pb(II) adsorption are shown in Figure 6 and Figure 7. We acquired each EDS spectrum at the 150 μm region. Pb signals in the cellulose were absent, which was likely related to the small amount of Pb(II) adsorbed, as shown in kinetic adsorption (see the next section). Pb(II) signals were mostly evident for all cellulose derivatives; for these scanned regions, Pb(II) content was higher for cellulose-chitosan derivatives (Com-1 and Com-3).

### 4.4. Kinetic Adsorption of Pb(II) by Functionalized Cellulose

Figure 8 shows that the adsorption capacity (*q_t_*) for all cellulose derivatives increased with time until a plateau region was obtained. During short contact times, there are likely many active sites available; thus, the adsorption process was highly favored. This was different in the plateau region; an equilibrium between active sites and Pb(II) ions in the solution was attained, and adsorption was not favored. Therefore, the adsorption capacity did not show a significant increase. This saturation of active sites was achieved after roughly four hours of contact for all probed materials. Experimental data were fitted to pseudo-first- and pseudo-second-order kinetic models. Results are shown in Figure 8 and Figure 9, as well as Table 2. For all samples studied, the pseudo-second-order kinetic model provided the best representation of the experimental data. This suggests that chemical reactions are predominant in the rate-controlling step, which agrees with the XPS results. Com-1, Com-3, and Com-4 *q_e_* values are only slightly different; these differences can be assigned to the interaction between Pb(II) and lone pairs of electrons in R–COO^−^ from succinate. In this case, the presence of amines and pyridines did not contribute significantly to Pb(II) adsorption.

### 4.5. Isothermal Adsorption of Pb(II) by Functionalized Cellulose

The Pb(II) adsorption by cellulose derivatives at different initial metal concentrations was tested, and experimental data were fitted to the Langmuir and Freundlich isotherm models. Results are shown in Figure 10 and Figure 11, as well as Table 3. In all cases, the Langmuir model was more suitable to describe the Pb(II) adsorption process. This model assumes a homogenous binding of the adsorbate at a monolayer surface, where *q_m_* and *K_L_* are constants related to the maximum capacity and metal ion affinity, respectively. These results are consistent with specific interactions between lone pair electrons and metal ions. Interestingly, *q_m_* values were 6.62, 43.14, 60.6, and 80.26 for cellulose, Com-1, Com-3, and Com-4, respectively. Naturally, the maximum adsorption capacity increased as cellulose was modified, obtaining the highest value for the cellulose modified with chitosan and pyridine. This behavior can be attributed to nitrogen lone pair electrons and the presence of chitosan and pyridine. Oxygen atoms were likely more reachable, and the interaction was favored at low Pb(II) concentrations. However, as the concentration of the metal ion increased, equilibrium conditions changed, and nitrogen began to interact with Pb(II), thereby increasing *q_m_*.

## 5. Conclusions

In summary, cellulose was modified by treating it with succinic anhydride, chitosan, and chitosan-picoly. This modification improved the adsorption capacity, especially at high Pb(II) concentrations. Cellulose derivatives adsorbed more lead than cellulose at pH 5 and room temperatures, increasing the maximum adsorption capacity from 6.62 to 80.26 mg/g of Pb(II) ions in an aqueous solution by batch adsorption. The improved adsorption capacity was presumably due to the cross-linking with chitosan. Additionally, incorporating the carbonyl group from succinic anhydride and methyl pyridine in the chitosan structure improved the absorption capacity. The methodology used in this experiment to study cellulose modification provides a new technique for incorporating different chelation groups; this could introduce derivatives that will efficiently remove heavy metal pollution.

## Data Availability

The data presented in this study are available on request from the corresponding author.

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
