# Peer review of "Chitosan Functionalized with 2-Methylpyridine Cross-Linker Cellulose to Adsorb Pb(II) from Water"

_polymers, 2021, doi:10.3390/polym13183166_

Round 1
Reviewer 1 Report
The manuscript submitted by Jorge Lozano-Montante et al reported the adsorption capacity of various functionalized celluloses for Pb(II) ions, It was found that the adsorption capacity of cellulose-chitosan-pyridine was the strongest, and the kinetic data analysis of the adsorption process showed a pseudo-second-order behavior. This manuscript has certain innovation and research significance. However, there are still some problems that need to be addressed in this manuscript in the current form. Here are my concerns:
Introduction
- This section needs major changes. There are too many descriptions about the hazards of heavy metal ions, but this is not the most important content. It is important to explain the method of adsorbing heavy metal ions and the advantages of cellulose and chitosan in this application, but this part content is less and needs to be supplemented.
- Page 1, line 35: “. 1 2” should be changed to “[1,2]”. There are many similar errors, please check the full text.
Materials and Testing Equipment
- Page 2, line 81: The authors did not provide a number of important characteristics of materials. The characteristics of cellulose used in the work along with CS molecular weight should be given.
- Page 3, line 111-112: “Preparation of Cellulose-Succinic-Chitosan (Com-3) and Cellulose-Succinic-Chi- 111 tosan-Pyridine (Com-4)” should be changed to “6. Preparation of Cellulose-Succinic-Chitosan (Com-3) and Cellulose-Succinic-Chi- 111 tosan-Pyridine (Com-4)”.
Results
- Page 10, “3. Surface analysis by SEM and EDS”, The SEM and EDS of cellulose and cellulose derivatives before Pb(II) adsorption should also be added to facilitate comparison.
- As the cost of sorbent is an important factor of application, it is necessary to consider its adsorption efficiency and preservation capacity. Therefore, the results of the adsorption cycle experiment should be added.
- The analysis in the article is relatively thin, and it is necessary to further study and explain the adsorption mechanisms of the material, for example, by comparing the infrared spectrum and the XPS spectrum before and after Pb(II) adsorption.
References
- Please check the format of the references, e.g., references “1. Kim, H. N.; Ren, W. X.; Kim, J. S.; Yoon, J., Fluorescent and colorimetric sensors for detection of lead, cadmium, and mercury ions. Chem Soc Rev 2012, 41 (8), 3210-3244.” should be corrected to “1. Kim, H. N.; Ren, W. X.; Kim, J. S.; Yoon, J., Fluorescent and colorimetric sensors for detection of lead, cadmium, and mercury ions. Soc. Rev. 2012, 41 (8), 3210-3244.”
- The authors could add the following references which would again increase the interest to general functional cellulosic material readers: Journal of Bioresources and Bioproducts, 2020, 5(4): 223-237; ACS Applied Materials & Interfaces, 2021, 13, 7617-7624; Journal of Bioresources and Bioproducts, 5(2): 79–95.
Author Response
Response to Reviewer 1 Comments
Introduction
- This section needs major changes. There are too many descriptions about the hazards of heavy metal ions, but this is not the most important content. It is important to explain the method of adsorbing heavy metal ions and the advantages of cellulose and chitosan in this application, but this part content is less and needs to be supplemented.
Response 1 Thank you for your comment. We make changes to the introduction as requested in the adsorption topic.
- Page 1, line 35: “. 1 2” should be changed to “[1,2]”. There are many similar errors, please check the full text.
Response 2 Thank you for your comment .All corrections were made in the document
Materials and Testing Equipment
- Page 2, line 81: The authors did not provide a number of important characteristics of materials. The characteristics of cellulose used in the work along with CS molecular weight should be given.
Response 3 Thank you for your comment. The information requested was placed in the document:
chitosan (CS) with a deacetylation degree ≥ of 75% and molecular weight around 150 kDa,
- Page 3, line 111-112: “Preparation of Cellulose-Succinic-Chitosan (Com-3) and Cellulose-Succinic-Chi- 111 tosan-Pyridine (Com-4)” should be changed to “6. Preparation of Cellulose-Succinic-Chitosan (Com-3) and Cellulose-Succinic-Chi- 111 tosan-Pyridine (Com-4)”.
Response 4Thank you for your comment. The change was done
Results
- Page 10, “3. Surface analysis by SEM and EDS”, The SEM and EDS of cellulose and cellulose derivatives before Pb(II) adsorption should also be added to facilitate comparison.
Response 5 Thank you for your comment. We believe that this study is not necessary since these EDS spectra were carried out in order to make qualitative comparisons between the adsorbed lead in each of the cellulose derivatives, clearly we would not obtain any signal for Pb(II) in cellulose derivatives that do not were in contact with lead
- As the cost of sorbent is an important factor of application, it is necessary to consider its adsorption efficiency and preservation capacity. Therefore, the results of the adsorption cycle experiment should be added.
Response 6 Thank you for your comment. We consider that the reviewer is right in his comment, however this suggestion is beyond the scope of this work.
- The analysis in the article is relatively thin, and it is necessary to further study and explain the adsorption mechanisms of the material, for example, by comparing the infrared spectrum and the XPS spectrum before and after Pb(II) adsorption.
Response 7 Thank you for your comment.
The XPS and FTIR comparison of the composite before and after lead adsorption was added to the article, as requested by the reviewer.
In addition, in the supplementary material section, the comparison of the rest of the composites by FTIR and an XPS table before lead adsorption is shown.
References
- Please check the format of the references, e.g., references “1. Kim, H. N.; Ren, W. X.; Kim, J. S.; Yoon, J., Fluorescent and colorimetric sensors for detection of lead, cadmium, and mercury ions. Chem Soc Rev 2012, 41 (8), 3210-3244.” should be corrected to “1. Kim, H. N.; Ren, W. X.; Kim, J. S.; Yoon, J., Fluorescent and colorimetric sensors for detection of lead, cadmium, and mercury ions. Soc. Rev. 2012, 41 (8), 3210-3244.”
Response 8 Thank you for your comment. The correction was done
- The authors could add the following references which would again increase the interest to general functional cellulosic material readers: Journal of Bioresources and Bioproducts, 2020, 5(4): 223-237; ACS Applied Materials & Interfaces, 2021, 13, 7617-7624; Journal of Bioresources and Bioproducts, 5(2): 79–95.
Response 9 Thank you for your comment. The correction was done

Reviewer 2 Report
In the submission, the authors provide a research study regarding chemical modification of the chitosan with 2-methylpyridine in order to increase the adsorption capacity. One comparative study regarding the adsorption capacity of cellulose, cellulose-succinic acid, cellulose-chitosan, and cellulose-chitosan-pyridine has been performed by the authors.
The kinetic data were analyzed to suggest one possible Pb(II) removal mechanism. Overall, this research provide important information regarding chemical modification of chitosan in order to obtain new materials with improved properties.
The topic of this research is very important for the scientific community due to the fact that Pb(II) is toxic to the aquatic environment and human health. Consequently, there is a need to develop new materials (biomaterials) with improved adsorption capacity to be used for heavy metals removal from wastewater.
The strong issues of the submission are:
- The title reflects the content of the paper.
- The key words are suitable so the article can be found in the current registers or indexes.
- The abstract is informative and completely self-explanatory.
- The introductory part presents an overview on current research on the issue investigated. The literature is sufficiently evaluated.
- The structure of the article is according to the structure of a research article.
- The presentation reflects the present state of knowledge.
- The authors explain how the experiments were performed. The article identifies the procedures followed. They are ordered in a meaningful way.
- The conclusions have been justified sufficiently.
- The text is presented in a manner that scientists in other disciplines will understand.
- The abbreviations and nomenclature are used according to the applicable international standards and rules.
- There are not any grammatical and spelling errors in the article that have to be corrected.
- The style of sentences and vocabulary is appropriate to the topic.
- The size of the article is appropriate to the content.
- The references are accurate and relevant for the subject of the paper.
- The authors acknowledge for the financial support.
However,
- I suggest to remove dot from the end of the title.
- I also suggest to add ”,” when listing bibliographic references. I sugest to replace ”14 15 16 17 18” with ”14-18” and ”31 32 33 34” with ”31-34”.
- I suggest to avoid using first person pronouns (''we') as well as personal experience in the paper.
- I suggest to rewrite the section ”2.6. Adsorption kinetics” to be more clear for the other readers. The kinetic study has to be performed at different times and the values of times should be mentioned.There is not sufficient information present for other researchers to replicate the research
- The authors should mention the repeatability of the tests performed and the standard deviation.
- I suggest to draw the possible structures of the Pb(II) loaded adsorbents based on the results of XPS analysis. FT-IR spectra of the the Pb(II) loaded adsorbents should be added to prove the removal mechanism and to confirm the XPS results.
- The authors should add desorption experiments to prove the reusability of the adsorbents.
- The authors should related their findings to other research results.
In my opinion, the paper analyzed addresses some important issues regarding the Pb(II) removal from wastewater.
The topic of this study is included in the important research fields of chemical engineering, environmental protection and environmental engineering, being relevant for environmental engineers.
This form of the paper can be published in the Processes after major changes.
Author Response
Response to Reviewer 2 Comments
- I suggest to remove dot from the end of the title.
Response 1: Thank you for your comment. The correction was done
- I also suggest to add ”,” when listing bibliographic references. I sugest to replace ”14 15 16 17 18” with ”14-18” and ”31 32 33 34” with ”31-34”.
Response 2 Thank you for your comment. The correction was done
- I suggest to avoid using first person pronouns (''we') as well as personal experience in the paper.
Response 3 Thank you for your comment. The correction was done
- I suggest to rewrite the section ”2.6. Adsorption kinetics” to be more clear for the other readers. The kinetic study has to be performed at different times and the values of times should be mentioned.There is not sufficient information present for other researchers to replicate the research
Response 4Thank you for your comment. The correction was done
- The authors should mention the repeatability of the tests performed and the standard deviation.
Response 5 Thank you for your comment. Each graph shown presents the average of three independent experiments and each measurement in each experiment was performed in triplicate. The values of qe, qm, etc were obtained by adjusting the averaged data that shown in these graphs, for that reason we could not obtain standard deviation values.
- I suggest to draw the possible structures of the Pb(II) loaded adsorbents based on the results of XPS analysis. FT-IR spectra of the the Pb(II) loaded adsorbents should be added to prove the removal mechanism and to confirm the XPS results.
Response 6 Thank you for your comment. The correction was done. We add the scheme 3, in this shows a possible mechanism of removal lead.
- The authors should add desorption experiments to prove the reusability of the adsorbents.
Response 7Thank you for your comment. We consider that the reviewer is right in his comment, however,this suggestion is beyond the scope of this work.
- The authors should related their findings to other research results.
Response 8Thank you for your comment. The correction was done.

Round 2
Reviewer 2 Report
All the comments are satisfactorily addressed by the authors. Consequently, the article is suitable for its publication in the Polymers journal.